# Metformin modulates the unfolded protein responses, altering lifespan and health-promoting effects in UPR-activated worms

Jerald Tan[1,¤a], Chutipong Chiamkunakorn[1*], Kanpapat Boonchuay[1,¤b], Yiying Shi[1], Bart P. Braeckman[2], Wichit Suthammarak[1*]

1 Department of Biochemistry, Faculty of Medicine Siriraj Hospital, Mahidol University, Bangkok, Thailand,
2 Department of Biology, Ghent University, Ghent, Belgium

☯ These authors contributed equally to this work.
¤a Current address: Department of Biochemistry, Yong Loo Lin-School of Medicine, National University of Singapore, Singapore, Singapore
¤b Current address: Akkhraratchakumari Veterinary College, Walailak University, Nakhon Si Thammara, Thailand
* wichit.sut@mahidol.edu (WS); pong.chutipong@gmail.com (CC)

## Abstract

Metformin has been demonstrated to extend lifespan in various model organisms, and its molecular effects are observed in the cytoplasm and multiple organelles, including mitochondria. However, its association with the unfolded protein response (UPR) and its impact on stress resistance and locomotion remain uncertain. In this study, metformin was found to exert differential influences on both UPR^mt and UPR^er. The correlation between metformin's lifespan-mediating effect and its interaction with UPRs was also inconsistent. We identified a metformin-mediated lifespan extension in wild-type *C. elegans* and in UPR^mt-activated *tomm-22* and *cco-1* RNAi worms. Metformin suppressed the UPR^mt without compromising the lifespan extension observed in *tomm-22* worms. Conversely, metformin did not affect the UPR^mt but extended the lifespan of long-lived *cco-1* RNAi worms. Furthermore, we investigated the effects of metformin on UPR^er-activated nematodes. We observed that metformin exhibited a slight increase in the UPR^er in *mdt-15* RNAi worms and failed to induce lifespan extension. Surprisingly, metformin appeared to mediate lifespan extension in *tmem-131* RNAi worms while suppressing the UPR^er. Notably, the correlation between thermotolerance, oxidative stress resistance, and the lifespan effects of metformin in UPR-activated worms was inconsistent. Activation of UPRs, but not metformin treatment, enhanced the locomotor phenotype of these worms.

## Introduction

Metformin is a biguanide compound commonly used to manage type 2 diabetes mellitus (T2DM). Notably, it exhibits lower toxicity compared to its predecessors,

**Data availability statement:** All relevant data are within the manuscript and its Supporting Information files.

**Funding:** Author who received the fund: WS - Funder and grant number: National Research Council of Thailand (NRCT) and Mahidol University (NRCT5-RSA63015-20) - URL of funder website: https://www.nrct.go.th/ - C. elegans strains were provided by the CGC, which is funded by NIH Office of Research Infrastructure Programs (P40 OD010440). ** The funders absolutely did not play any role in the study design, data collection and analysis, decision to publish, or preparation of the manuscript.**

**Competing interests:** The authors have declared that no competing interests exist.

**Abbreviations:** UPRmt, mitochondrial unfolded protein response; UPRer, endoplasmic reticulum unfolded protein response; GFP, green fluorescent protein; T2DM, type 2 diabetes mellitus.

phenformin and buformin [1]. Beyond its primary role in treating T2DM, studies suggests that metformin may reduce the risk of T2DM-related diseases, including dementia [2], cardiovascular disease [3] and cancer [4]. Furthermore, longevity studies have demonstrated an extension of lifespan with a reduction of age-related symptoms in animal models administered metformin at various doses [5–9], implying that metformin not only possesses pro-longevity properties but also exhibits health-promoting effects. Several pathways have been identified as mediating metformin-induced longevity, including activation of the lysosomal pathway [7], inhibition of mitochondrial complex I [8,10,11], inhibition of adenylate cyclase [12], alteration of folate metabolism [13], direct regulation of AMP-kinase [14–16], and enhancement of autophagy [17].

Several of these pathways are associated with the mitochondria, where positively charged molecules such as metformin may accumulate [18–21]. These associations led us to consider the possibility that the mitochondrial unfolded protein response (UPR$^{mt}$) may have a role to play in metformin-mediated lifespan extension, stress resistance and health promotion. In *C. elegans*, the UPR$^{mt}$ is a mitochondrial stress response that can extend lifespan by preserving mitochondrial transport [22,23] in a cell-non-autonomous manner [24]. Upon mitochondrial proteomic stress, the UPR$^{mt}$ signaling pathway is activated, which initiates the translocation of ATFS-1, a transcription factor, into the nucleus. ATFS-1, in conjunction with other proteins such as UBL-5 and DVE-1 regulates the expression of genes essential for mitigating the effects of proteomic stress [25–27]. Another similar stress response is the unfolded protein response for the endoplasmic reticulum (UPR$^{er}$), which can be activated by proteomic stress events within the endoplasmic reticulum. In addition to stress response to proteomic toxicity in the endoplasmic reticulum, UPR$^{er}$ is essential for normal development of *C. elegans* [28–30]. However, UPR$^{mt}$ may not be required for several types of lifespan extension in the nematode, and UPR$^{mt}$ *per se* does not predict longevity in the nematodes [31,32]. Interestingly, the effect of metformin on UPR$^{mt}$ has only been briefly discussed previously, stating only that metformin itself does not activate UPR$^{mt}$ [8]. Furthermore, little is known about the interaction of metformin in worms exhibiting a short-lived phenotype, whether it can display a beneficial effect on their lifespan, stress resistance, and health.

In this study, we investigated the effect of metformin on lifespan, its interaction with UPR$^{mt}$, UPR$^{er}$, and it influence on health promotion (stress resistance and locomotion) in UPR-activated *C. elegans*. We hypothesized that metformin may mediate its beneficial effect in the nematode through UPR$^{mt}$, but not UPR$^{er}$. To investigate UPR$^{mt}$, we employed *tomm-22* and *cco-1* RNAi knockdowns, both constitutively activating UPR$^{mt}$ but with opposing effects on worm lifespan [32]. TOMM-22 is involved in protein transport across the outer mitochondrial membrane, while CCO-1 serves as a subunit of the respiratory complex IV. For UPR$^{er}$, we employed *mdt-15* and *tmem-131* RNAi knockdowns, both resulting in activated UPR$^{er}$ [33–35] and shortened lifespan. MDT-15 is a mediator subunit required for transcription of genes involved in fatty acid metabolism [36,37], while TMEM-131 is a chaperone in the ER and is essential for collagen recruitment and secretion [34].

Here, we report that metformin induces lifespan extension and mildly increases thermotolerance and oxidative stress resistance in middle-aged wild-type worms. In the UPR-activated worms, metformin extended lifespan in *tomm-22*, *cco-1* and *tmem-131* worms while suppressing UPR$^{mt}$ in *tomm-22* but not *cco-1*. Metformin reduced the lifespan of *mdt-15* worms and slightly increased their UPR$^{er}$. Notably, metformin suppressed the UPR$^{er}$ in *tmem-131* while simultaneously extending the worms' lifespan. Stress resistance in metformin-treated UPR-activated worms deviated from the wild-type patterns and was not directly correlated with lifespan. Activation of UPRs resulted in enhanced locomotion, particularly in *tomm-22*. However, metformin did not exhibit any beneficial effect on locomotion in the nematodes. These findings indicate that metformin exhibits a multifaceted interaction with lifespan, UPR activation, and stress resistance, but not locomotory function in *C. elegans*.

## Materials and methods

### Nematode strains and bacteria cultures

Nematode strains were obtained from the Caenorhabditis Genetics Centre (CGC, University of Minnesota, USA), including: Wild-type N2 Bristol, SJ4100 zcls13[*hsp-6*::GFP], SJ4005 zcls4[*hsp-4*::GFP]. *E. coli* strains from Ahringer's Library used include OP50, HT115 harboring the empty vector L4440 (EV), *tomm-22* (W10D9.5), *cco-1* (F26E4.9), *nuo-6* (W01A8.4), *mdt-15* (R12B2.5), *tmem-131* (C27F2.8), *tag-335*(C42C1.5), *atfs-1*(ZC376.7), and *ubl-5*(F46F11.4)

### Nematode culture and RNA interference

Nematode Growth Media (NGM) was prepared as described [38], with the volume adjusted to accommodate the addition of metformin to molten agar. The final concentrations of metformin used were 25 mM and 50 mM. The cooled plates were subsequently seeded with *E. coli* and incubated overnight at 37 ºC in an incubator. RNA interference by feeding was performed as described [39].

### Lifespan analysis

Wild-type N2 worms were synchronized by transferring egg-laying day-1 adult worms to freshly seeded EV or OP50 for 4−6 hours. Subsequently, they were removed from the plates and the plates were incubated at 20 ºC for 3 days before late L4 worms were picked to individual plates (20−30 per plate) for maintenance and data collection. Worms were synchronized at a well-fed state for at least two generations prior to experimentation. RNAi bacteria were utilized to knock down the genes *tomm-22*, *cco-1*, *nuo-6*, *mdt-15*, *tmem-131*, *tag-335, atfs-1*, and *ubl-5*. Worms were transferred to fresh plates daily for the initial 10 days, followed by every other day thereafter. The presence of life was determined by touch provocation and pharyngeal pumping. Worms exhibiting vulvar deformities, desiccation, or bagging were censored and excluded from the analysis.

### Green fluorescent protein image capturing and signal quantitation

An inverted fluorescence microscope equipped with a camera (Ti-S Intensilight Ri1 NIS-D) (Nikon, Tokyo, Japan) was used to visualize GFP fluorescence under the FITC Filter (Excitation 465–495 nm, Emission at 515–555 nm). The NIS-D Elements software was used to capture the images in.tif format.

For quantification of worm fluorescence, the mean pixel density of individual worms was calculated using Fiji (Version 2.1.0). Regions of interests (ROIs) were drawn using the polygonal selection tool around each individual worm (n = 5) in the brightfield image and saved in the ROI manager for superimposition in the fluorescence image. Then, using the same ROI in the brightfield image, the fluorescence intensity of each individual worm was quantified. The background normalization signal was calculated by manual thresholding of the blue channel, binarization, then filling holes and erosion-- dilation to select the worms before using inverse selection for the area outside of the worms.

The final fluorescence was then calculated by deducting the individual worm fluorescence with the background signal, and the mean and standard deviation was calculated and plotted. Student *t*-test was used for statistical analysis.

## Thermotolerance and oxidative stress assays

Wild-type N2 worms were synchronized in a manner similar to the lifespan experiments conducted prior to the commencement of the stress experiments, with the addition of 30 μM FUdR. Subsequently, worms were transferred to 96-well plates containing 100 μl S-basal every 3 days for analysis (n = 10 per well, 3 wells per strain per condition). SYTOX Green was added to a final concentration of 5 μM prior to the commencement of the experiment. For thermotolerance, worms were exposed to 42°C for 1 hour before manual scoring for death (using a fluorescence microscope equipped with a FITC Filter) every subsequent hour until all the worms were dead. For oxidative stress, worms were exposed to 1.4% tert-Butyl peroxide for 2 hours at 20°C before manual scoring every hour until all the worms were dead.

## Locomotion assays

Wild-type N2 worms were cultured synchronously on EV and different RNAi bacteria including *tomm-22* and *mdt-15*, and tested on the 3$^{rd}$, 6$^{th}$, 9$^{th}$ and 12$^{th}$ day of growth at 20°C. Locomotion assays included body trashing and crawling velocity measurements. In a trashing assay, 10–20 worms were placed into 200 μl of M9 solution. The worms were allowed to equilibrate in the solution for 1 minute before the trashing movie of the worms was recorded for 30 seconds using a digital camera mounted onto a stereomicroscope. In a crawling velocity assay, 10–20 worms were transferred onto an unseeded NGM plate and allowed to crawl freely for 1 minute before the movement movie of the worms was recorded for 30 seconds. Each video ensured worms moved independently, without overlap, and under uniform illumination. The videos were exported and analyzed using ImageJ (Fiji) (Version 2.1.0). The data were collected from 3 independent cultures of each worm strain.

## Statistical analysis

The lifespan results were analyzed using the log-rank test implemented in OASIS [40]. Thermotolerance, oxidative stress, trashing and velocity assays were analyzed by either Student *t*-test or the log-rank test.

## Results

### The lifespan-extending effects of metformin are not consistently observed in all UPR-activated worm strains

We observed a significant lifespan extension (p < 0.01) in wild-type worms treated with 50 mM metformin (Fig 1A). The nematodes did not exhibit consistent lifespan responses to RNAi-activated UPR. UPR$^{er}$ activation by *mdt-15* and *tmem-131* (RNAi) reduced the nematode's lifespan, while UPR$^{mt}$ activation led to lifespan extension in *cco-1* (RNAi) and a reduction in *tomm-22* (RNAi) (Fig 1B). Interestingly, exposing metformin to the UPR-activated worms also yielded varying results. For UPR$^{mt}$-activated worms, metformin treatment at 50 mM in short-lived *tomm-22* and long-lived *cco-1* worms resulted in a significant lifespan extension when compared to the corresponding strain that was not treated with metformin (p < 0.01) (Fig 1C and 1D). Conversely, the UPR$^{er}$-activated short-lived *mdt-15* worms exhibited a significant lifespan reduction upon metformin treatment at 50 mM when compared to *mdt-15* worms that were not treated with metformin (p < 0.01) (Fig 1E). Metformin treatment of UPR$^{er}$-activated short-lived *tmem-131* worms significantly extended the lifespan of the animals compared to *tmem-131* worms that were not treated with metformin (p < 0.05) (Fig 1F). In summary, metformin extends lifespan of UPR$^{mt}$-activated *tomm-22* and *cco-1*, and UPR$^{er}$-activated *tmem-131* but not UPR$^{er}$-activated *mdt-15*. Statistical analysis details are presented in Table 1.

A previous study by Espada *et al.* [41] reported that metformin-mediated longevity *via* UPR$^{mt}$ required *ubl-5*. To further distinguish whether the lifespan-extending effect of metformin was UPR$^{mt}$-dependent in the *tomm-22* background, we investigated the effect of *ubl-5* in the *tomm-22* background. Logically, if *ubl-5* was required for metformin-mediated longevity, then the effect of metformin would be lost in the presence of the *tomm-22* background. Surprisingly, we found that the double knockdown of *tomm-22* and *ubl-5* in the presence of metformin did not abrogate the lifespan-extending effects of

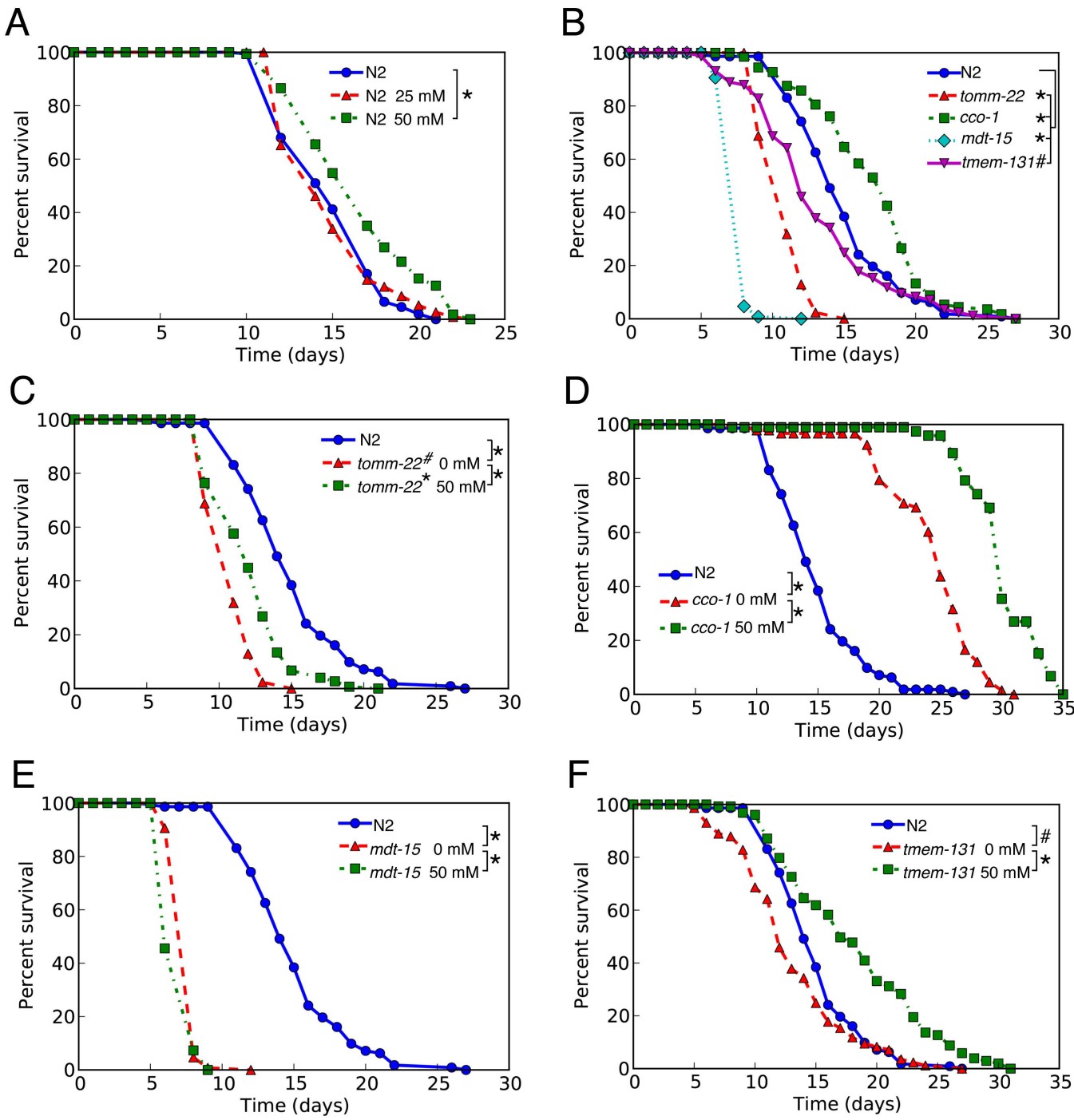

**Fig 1. Effect of metformin on lifespan of wild-type and UPR-activated *C. elegans*.** Lifespan assays were conducted in N2 wild-type nematodes under different conditions. The nematodes were grown on OP50 in the presence of 25 mM or 50 mM metformin or without metformin (A). Nematodes were grown on either EV or RNAi bacteria for UPRmt activation (*tomm-22* and *cco-1*) or UPRer activation (*mdt-15* and *tmem-131*) without metformin treatment (B). The lifespan of UPR-activated nematodes in the presence of 50 mM metformin was assayed in comparison to the control, which is N2 grown on EV, as shown in (C-F). Each assay consisted of 100-200 age-synchronized worms. Statistical analysis was performed using the log-rank test. * and # denote statistical significance where Bonferroni's p-value < 0.01 and < 0.05, responsively. Statistical analysis details are presented in Table 1.

**Table 1. Statistical output of Log-Rank test for lifespan of wild-type and UPR-activated worms.**

| Strain | Metformin | n | Lifespan | | | p-value | Bonferroni's p-value | p-value condition |
|---|---|---|---|---|---|---|---|---|
| | | | Mean | Median | Max | | | |
| N2 fed OP50 | 0 mM | 153 | 14.98±0.21 | 15 | 21 | | | |
| | 25 mM | 115 | 14.89±0.27 | 14 | 23 | 0.8702 | 1.0000 | vs N2 fed OP50 0 mM |
| | 50 mM | 156 | 16.60±0.29 | 17 | 23 | 5.3E-07 | **1.1E-06**[*] | |
| N2 fed EV | 0 mM | 150 | 14.92±0.32 | 14 | 27 | | | |
| tomm-22 (UPRmt) | 0 mM | 173 | 10.87±0.12 | 11 | 15 | 0.0E+00 | **0.0E+00**[*] | vs N2 fed EV 0 mM |
| | 50 mM | 165 | 12.17±0.20 | 12 | 21 | 1.1E-08 | **2.3E-08**[*] | vs tomm-22 0 mM |
| cco-1 (UPRmt) | 0 mM | 100 | 24.32±0.46 | 25 | 31 | 0.0E+00 | **0.0E+00**[*] | vs N2 fed EV 0 mM |
| | 50 mM | 100 | 29.97±0.42 | 30 | 35 | 0.0E+00 | **0.0E+00**[*] | vs cco-1 0 mM |
| mdt-15 (UPRer) | 0 mM | 128 | 7.88±0.06 | 9 | 12 | 0.0E+00 | **0.0E+00**[*] | vs N2 fed EV 0 mM |
| | 50 mM | 123 | 6.98±0.10 | 8 | 9 | 0.0E+00 | **0.0E+00**[*] | vs mdt-15 0 mM |
| tmem-131 (UPRer) | 0 mM | 150 | 13.06±0.45 | 12 | 27 | 0.0075 | **0.0151**[#] | vs N2 fed EV 0 mM |
| | 50 mM | 150 | 18.16±0.53 | 17 | 31 | 0.0E+00 | **0.0E+00**[*] | vs tmem-131 0 mM |

Statistical analysis was conducted using the log-rank test.

[*]and

[#]denote statistical significance where Bonferroni's p-value<0.01 and < 0.05, responsively.

metformin but further synergized to extend lifespan (S1 Fig). These findings suggest that both *tomm-22* and *ubl-5* do not overlap in their pathways, supporting our hypothesis that metformin's effect in longevity is independent of UPRmt.

## Metformin exposure affected both UPRmt and UPRer

We next investigated whether metformin exhibits differential interactions with both UPR systems. To this end, we employed the GFP-tagged reporter proteins *hsp-6*::GFP and *hsp-4*::GFP to observe the effect of metformin on the activation of the UPRmt and UPRer, respectively.

Our findings revealed that *cco-1*(RNAi);*hsp-6*::GFP worms exhibited no discernible differences in fluorescence profiles between metformin-treated and non-treated groups (Fig 2A–B). When *tomm-22*(RNAi);*hsp-6*::GFP and *tmem-131*(RNAi);*hsp-4*::GFP worms were subjected to 50 mM metformin, a decrease in the UPRmt was observed. In addition, metformin appeared to enhance UPRer, slightly but significantly, in *mdt-15*(RNAi);*hsp-4*::GFP (Fig 2A–B). To enhance the comprehensiveness of our analysis regarding the interaction between metformin and the UPRs, we also tested additional UPR-activated worms. These included *nuo-6* RNAi (UPRmt) and *tag-335* RNAi (UPRer) worms, whose lifespan and stress resistance phenotypes, however, were not carried on as in others. We found that metformin suppressed UPRmt in *nuo-6*(RNAi);*hsp-6*::GFP while increased UPRer in *tag-335*(RNAi);*hsp-4*::GFP. (Fig 2A–B).

These findings indicate that metformin's action is not mitochondria-specific, as it can influence both the UPRmt and UPRer pathways. Furthermore, they imply that UPRmt is dispensable for lifespan extension, as the metformin-induced reduction of UPRmt (Fig 2) did not diminish its lifespan-extending effect in *tomm-22* (Fig 1C). It is then plausible that UPRmt can be inhibited by metformin. Similarly, these results indicate that UPRer is not necessary for the lifespan-extending effect of metformin, as *tmem-131* exhibited extended lifespan upon metformin treatment despite decreasing UPRer (Figs 1F and 2).

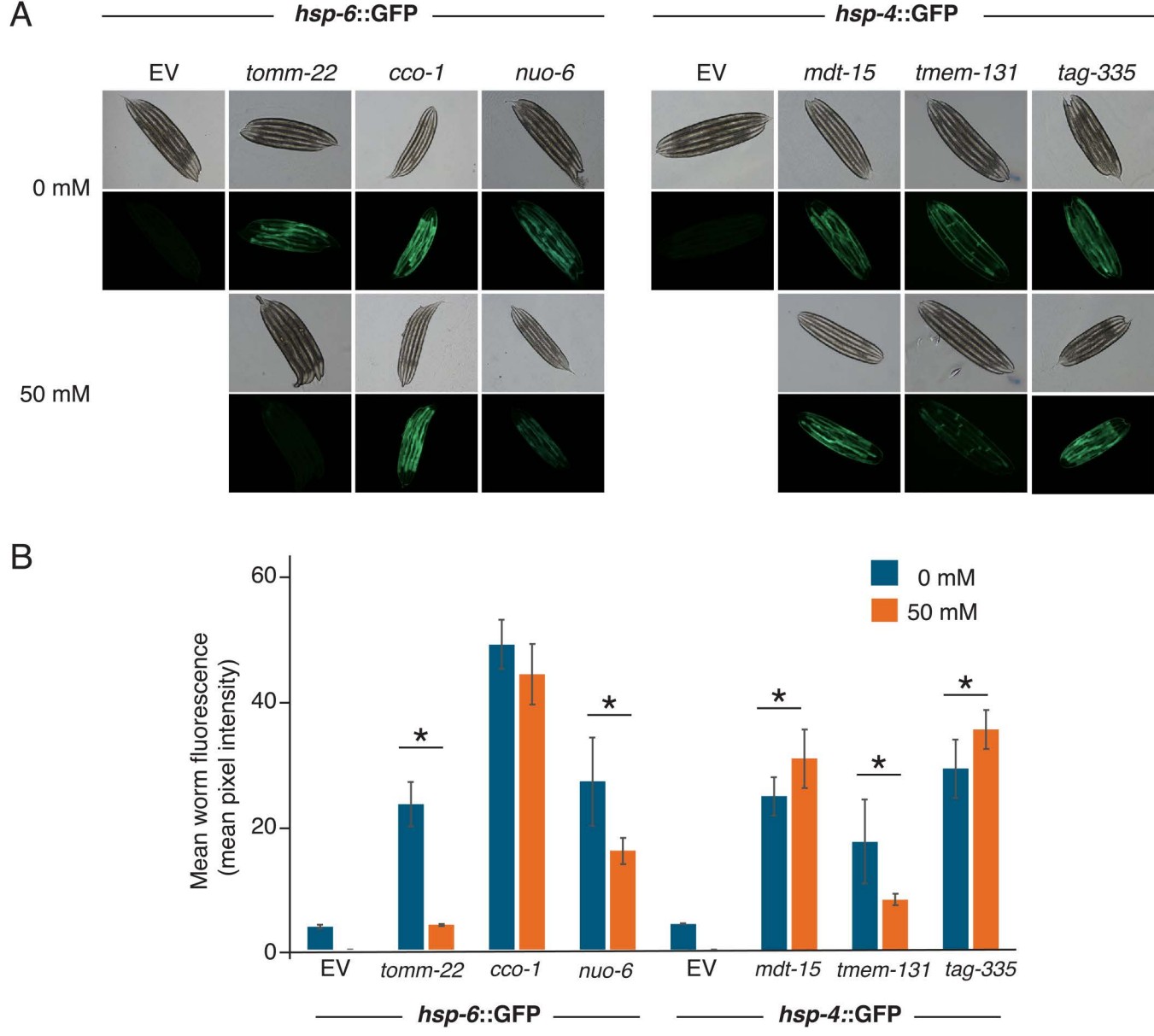

**Fig 2. Effect of metformin on UPRs activation.** Expression profiles of UPR^mt and UPR^er were conducted in SJ4100 zcls13[*hsp-6*::GFP] and SJ4005 zcls4[*hsp-4*::GFP] worms, accordingly. The expression of *hsp-6*::GFP in worms fed with EV, *tomm-22*, *cco-1* and *nuo-6* RNAi, and of *hsp-4*::GFP in worms fed with EV, *mdt-15*, *tmem-131* and *tag-355* RNAi is presented as shown (A). The treated nematodes were administered metformin at a concentration of 50 mM. Quantitative analysis of GFP signals is presented in (B). The assay was conducted on adult day 1 worms. (n = 5 for each experiment).* denotes p-value < 0.05.

## Metformin enhances thermotolerance in young wild-type adults, but not in UPR-activated nematodes

We showed that metformin extended the lifespan in *tomm-22* worms while simultaneously suppressing the activated UPR^mt, a mechanism that mitigates mitochondrial stress. We next investigated the effects of metformin on the resistance to two stressors: heat and oxidative stress.

Young metformin-treated wild-type worms showed moderately but significantly enhanced thermotolerance, a characteristic that diminished from Day 9 of life. Metformin did not alter thermotolerance throughout the entire adult life in both *tomm-22* and *mdt-15* (Fig 3B,C, Table 2). It is noteworthy that the mean thermotolerance of untreated *mdt-15* surpassed 10 hours, comparable to that of 50 mM metformin-treated wild-type worms but lower than that of the untreated ones (Table 2). Therefore, UPR^er activation may enhance thermotolerance in worms, but metformin treatment does not further augment this effect.

In summary, metformin treatment resulted in increased thermotolerance only in young wild-type adults. Activation of UPR^mt and UPR^er by knocking down *tomm-22* and *mdt-15* caused young metformin-treated worms to lose their enhanced thermotolerance. This outcome is somewhat surprising, as the induced UPR would be anticipated to protect the worms against heat stress. Notably, while mean heat stress survival is not statistically different, metformin occasionally induces a slightly stronger shoulder but steeper decline.

### Metformin promotes late-life oxidative stress resistance in UPR^mt-activated *tomm-22* nematodes, but not in UPR^er-activated *mdt-15* nematodes

Wild-type worms given metformin exhibited enhanced oxidative stress resistance in Day-6 worms (Fig 4A, Table 3). In contrast, the UPR^mt-activated worms displayed a more intricated response. Although oxidative stress resistance in

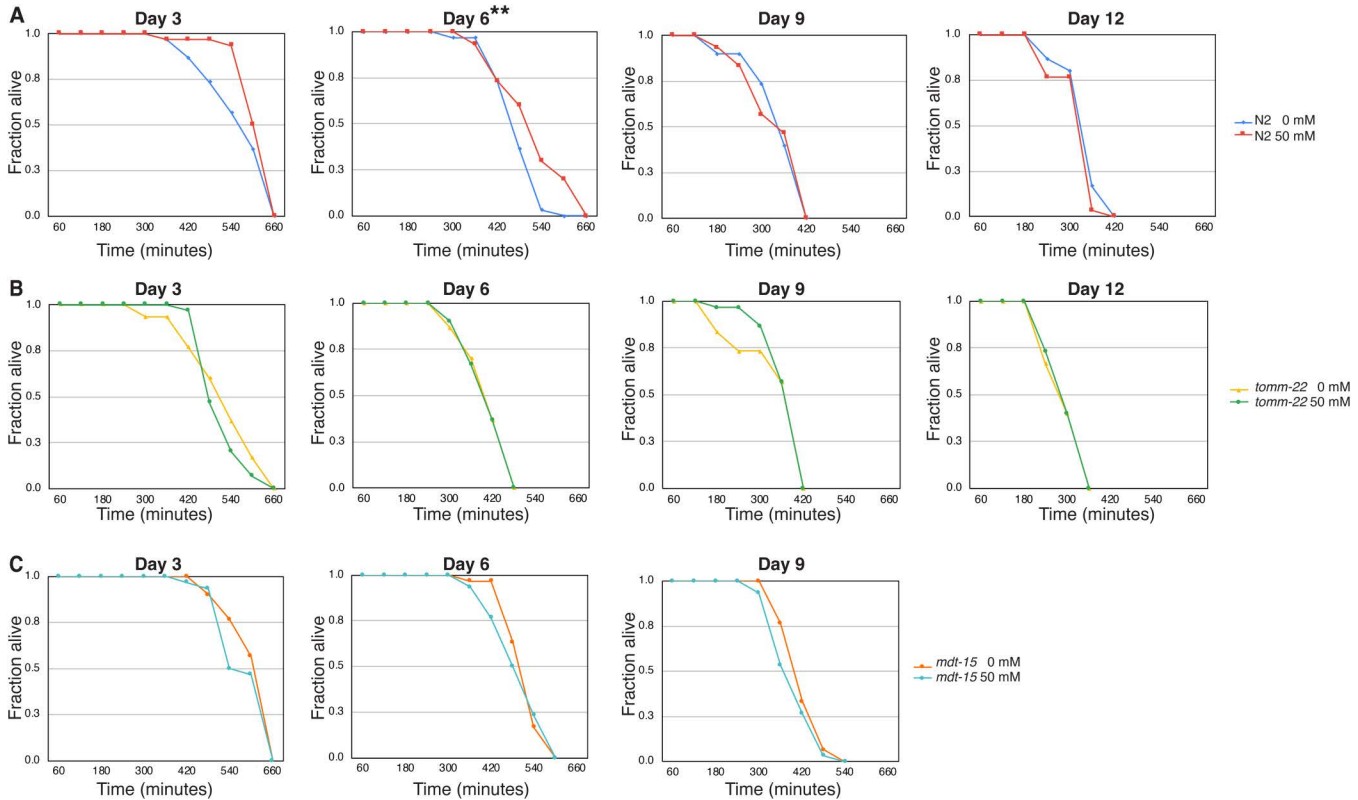

**Fig 3. Effect of metformin on thermotolerance in wild-type and UPR-activated *C. elegans*.** Thermotolerance assays were performed on N2 wild-type worms fed with EV (A), *tomm-22* RNAi (B), and *mdt-15* RNAi (C). Worms were exposed to metformin for their entire life (0 mM and 50 mM, as indicated). The assay was conducted every 3 days. (n = 30 for each experiment). * denotes p-value < 0.05. Statistical analysis details are presented in Table 2.

**Table 2. Statistical output of overall and mean thermotolerance survival of wild-type and UPR-activated worms.**

| Worm | Age | Overall Thermotolerance Survival | | Mean Thermotolerance Survival | | |
|---|---|---|---|---|---|---|
| | | p-value | Bonferroni's p-value | 0 mM Metformin (min) | 50 mM Metformin (min) | p-value (t-test) |
| N2 fed EV | Day 3 | 0.0503 | 0.0503 | 570 ± 16.43 | 620 ± 10.71 | **0.0135**\* |
| | Day 6 | 0.0141 | **0.0141**\* | 484 ± 10.93 | 526 ± 16.90 | **0.0413**\* |
| | Day 9 | 0.9825 | 0.9825 | 356 ± 13.25 | 348 ± 14.25 | 0.6825 |
| | Day 12 | 0.1945 | 0.1945 | 350 ± 9.42 | 334 ± 9.67 | 0.2408 |
| tomm-22 (UPR^mt) | Day 3 | 0.3480 | 0.3480 | 526 ± 18.04 | 522 ± 10.06 | 0.8492 |
| | Day 6 | 1.0000 | 1.0000 | 416 ± 11.29 | 416 ± 10.93 | 1.0000 |
| | Day 9 | 0.0961 | 0.0961 | 352 ± 17.14 | 382 ± 9.99 | 0.1359 |
| | Day 12 | 0.8591 | 0.8591 | 304 ± 9.35 | 308 ± 8.82 | 0.7568 |
| mdt-15 (UPR^er) | Day 3 | 0.3707 | 0.3707 | 614 ± 11.20 | 592 ± 12.56 | 0.1910 |
| | Day 6 | 0.5981 | 0.5981 | 524 ± 9.35 | 506 ± 13.17 | 0.2697 |
| | Day 9 | 0.0961 | 0.0961 | 430 ± 9.42 | 406 ± 10.84 | 0.1001 |

Log Rank analysis was used to calculate statistical significance of overall thermotolerance survival, comparing survival rates between 0 mM and 50 mM metformin.

\*denotes statistical significance where Bonferroni's p value or Student t-test p-value < 0.05.

metformin-treated *tomm-22* worms was decreased in young Day-3 worms, it was significantly increased during late life at Days 9 and 12 (Fig 4B, and Table 3).

It is noteworthy that untreated Day-3 *tomm-22* worms survived the oxidative stress for 610 minutes while the corresponding wildtypes survived only for 522 mins (Table 3). This increased oxidative stress resistance in *tomm-22* worms declined with age (Table 3). Furthermore, the enhanced oxidative stress resistance observed at early age was obliterated by metformin. The mean oxidative stress resistance of Day-3 *tomm-22* worms decreased from 610 minutes (untreated) to 536 minutes (50 mM metformin), which is close to the untreated Day 3 wildtypes (522 minutes) (Table 3). The potential reversal of this effect in older worms may be attributed to the extended healthspan effect of metformin in *tomm-22* worms.

The activation of UPR^er by *mdt-15* RNAi resulted in a slight increase in oxidative stress resistance in young to middle-aged worms compared to the wildtype (Fig 4C, and Table 3). Conversely, metformin caused a slight but significant reduction in mean survival time in oxidative stress resistance in Day-6 worms (Table 3).

### UPR activation, but not metformin, has a beneficial effect on locomotor phenotypes in the nematodes

Next, we investigated whether metformin could exhibit a beneficial effect on locomotion of the worms by measuring body trashing and crawling velocity in N2 wild-type and UPR-activated nematodes. Without metformin, UPR^mt-activated nematode *tomm-22* exhibited significantly higher locomotory rates (both trashing and crawling velocity) compared to N2 wild-type worms (Fig 5A and 5E) from Day 6 to late life. Nematodes with UPR^er activated by *mdt-15* exhibited significantly lower trashing rates in D3 but higher trashing rates in D6, compared to N2. When metformin was supplemented, N2 wild-type worms showed significantly slower trashing rates in older worms (Fig 5B) while crawling velocity remained unchanged (Fig 5F). In metformin-treated UPR-activated worms, neither *tomm-22* nor *mdt-15* exhibited any alteration in trashing or crawling velocity for the most extensively tested lifetime, with the exception of *mdt-15* at D9, which displayed a significantly elevated trashing rate (Fig 5C–D and 5G–H).

### Discussion

For several decades, mitochondria have been considered the primary driver of the aging process [42] and the UPR^mt has been associated with increased longevity in several model organisms [43–47]. This prompted us to

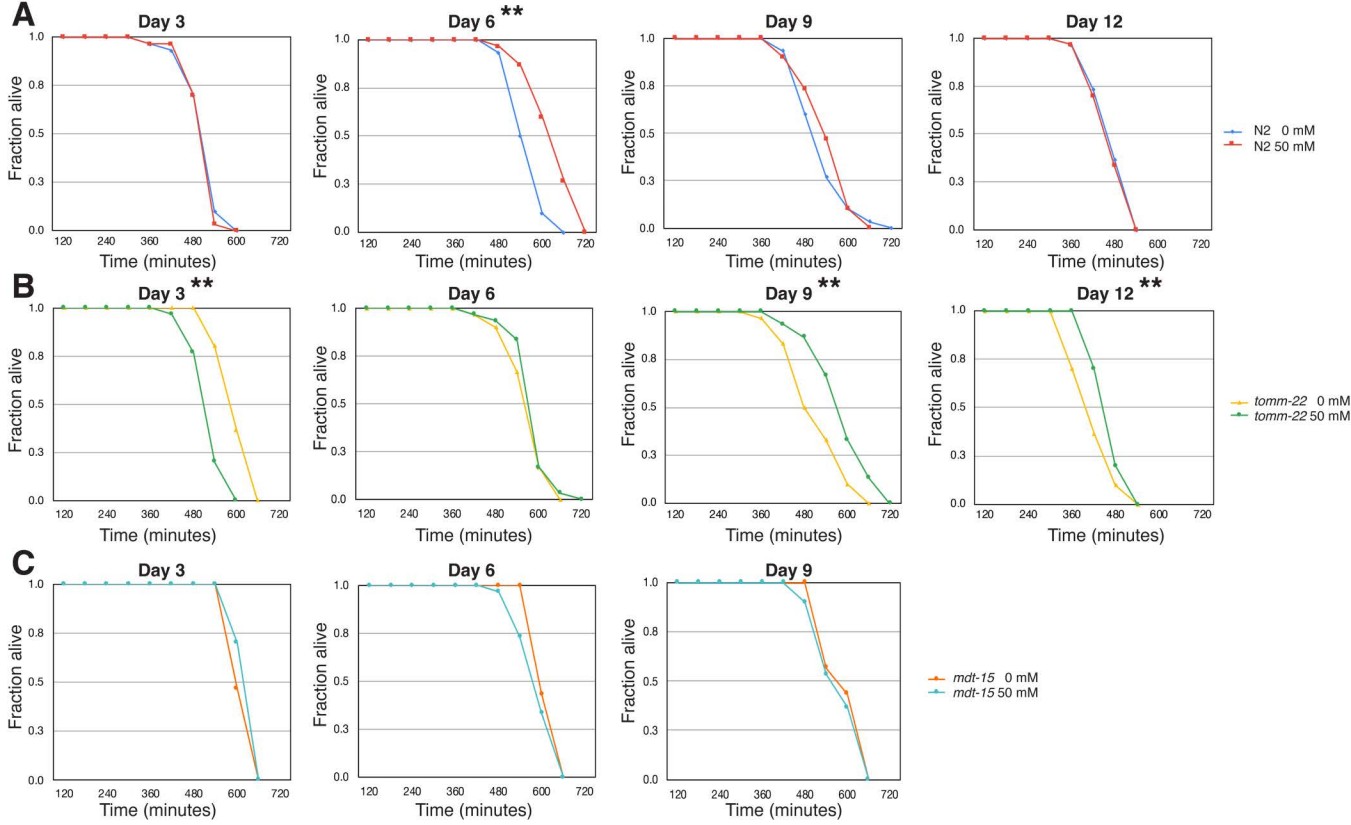

**Fig 4. Effect of metformin on oxidative stress resistance in wild-type and UPR-activated *C. elegans*.** Oxidative stress assays were performed on N2 wild-type worms fed with EV (A), *tomm-22* RNAi (B), and *mdt-15* RNAi (C). Worms were treated with metformin for their entire life (0 mM and 50 mM, as indicated). The assay was conducted every 3 days. (n = 30 for each experiment). ** denotes p-value < 0.01. Statistical analysis details are presented in Table 3.

investigate whether UPR^mt, a biological response that specifically designed to counteract mitochondrial stress, can be modulated by metformin, a compound that is known to interact with mitochondrial pathways and exhibits anti-aging properties.

In this study, we demonstrated that metformin extends the lifespan of wild-type worms, aligning with previous studies findings [6–8,13]. The use of 50 mM metformin for nematodes appears to be sufficient to induce the longevity phenotype. However, a higher dosage may be necessary for polar drugs, as the protective cuticle of the nematode can restrict their uptake. Nevertheless, metformin has also been shown to reduce lifespan in some circumstances, primarily due to overstimulation of the phenotype or drug toxicity [48,49]. We demonstrated that the pro-longevity effect of metformin is maintained in both *tomm-22* (membrane transport associated-UPR^mt) and *cco-1* (electron transport associated-UPR^mt). Furthermore, by investigating *ubl-5;tomm-22* double RNAi knockdown, we found that the lifespan extending effect by metformin was not lost but further synergized to extend lifespan. Interestingly, we also demonstrated that metformin extends the lifespan of *tmem-131* (protein sorting associated-UPR^er) while decreasing UPR^er. Conversely, metformin decreased the lifespan of *mdt-15* (modulator associated-UPR^er) while increasing UPR^er. These findings suggest that metformin's beneficial effect on worm lifespan is independent of UPR^mt or UPR^er activation. We propose that metformin may have some underlying interplay between various components in the unfolded stress response systems and other pathway, secondary to the target gene knockdown.

**Table 3. Statistical output of overall and mean oxidative stress resistance of wild-type and UPR-activated worms.**

| Worm | Age | Overall Oxidative Stress Resistance | | Mean Oxidative Stress Resistance | | |
|---|---|---|---|---|---|---|
| | | p-value | Bonferroni's p-value | 0 mM Metformin (min) | 50 mM Metformin (min) | p-value (t-test) |
| N2 fed EV | Day 3 | 0.6816 | 0.6816 | 522 ± 9.01 | 520 ± 7.66 | 0.8663 |
| | Day 6 | 9.8E-06 | **9.8E-06**\*\* | 512 ± 8.36 | 582 ± 11.71 | **0.0001**\*\* |
| | Day 9 | 0.3801 | 0.3801 | 536 ± 12.63 | 552 ± 12.46 | 0.3709 |
| | Day 12 | 0.7590 | 0.7590 | 484 ± 9.35 | 480 ± 9.38 | 0.7637 |
| tomm-22 (UPR\textsuperscript{mt}) | Day 3 | 2.0E-07 | **2.0E-07**\*\* | 610 ± 8.04 | 536 ± 7.97 | **0.0001**\*\* |
| | Day 6 | 0.2802 | 0.2802 | 582 ± 10.26 | 596 ± 9.77 | 0.3272 |
| | Day 9 | 0.0014 | **0.0014**\*\* | 524 ± 14.40 | 596 ± 14.68 | **0.0009**\*\* |
| | Day 12 | 0.0031 | **0.0031**\*\* | 428 ± 10.09 | 474 ± 7.67 | **0.0006**\*\* |
| mdt-15 (UPR\textsuperscript{er}) | Day 3 | 0.0691 | 0.0691 | 568 ± 5.47 | 582 ± 5.02 | 0.0643 |
| | Day 6 | 0.0904 | 0.0904 | 626 ± 5.43 | 602 ± 9.16 | **0.0280**\* |
| | Day 9 | 0.4681 | 0.4681 | 540 ± 10.20 | 528 ± 11.45 | 0.4371 |

Log Rank analysis was used to calculate statistical significance of overall oxidative stress resistance comparing 0 mM and 50 mM metformin. Statistical significance was denoted by asterisks

(\*) and double asterisks

(\*\*) when Bonferroni's p-value or Student t-test p-value < 0.05 and < 0.01, respectively.

We had initially expected that metformin may be beneficial for short-lived phenotypes by exerting its lifespan-extending effect through UPR\textsuperscript{mt}. Nonetheless, we demonstrated that metformin did not activate UPR\textsuperscript{mt}, consistent with previous studies (8, 51). In addition, we presented evidence suggesting that metformin was able to significantly reduce UPR\textsuperscript{mt} activation in tomm-22 and nuo-6 worms, but not in cco-1. This might be attributed to the negative effect of tomm-22 RNAi on mitochondrial import as recently demonstrated by Xin, et al. [22]. Upon activation of UPR\textsuperscript{mt} in tomm-22, mitochondrial import was also affected, potentially leading to degraded stress-responsive signals emanating from the nucleus. Conversely, mitochondrial import was found to be enhanced in cco-1, which can attenuate stress response. Therefore, we hypothesize that UPR\textsuperscript{mt} suppression by metformin can outcompete the stress-responsive signals from the nucleus to restore proteostasis in tomm-22 but not in cco-1. Interactions between metformin and other UPR\textsuperscript{mt} effectors such as HSP-6 or ATFS-1 maybe unlikely, as metformin has no inhibitory effect on the cco-1-induced UPR\textsuperscript{mt}. In addition, the primary role of tomm-22 is a mitochondrial outer membrane transporter rather than a mitochondrial UPR stress regulator. The activation of hsp-6 response in tomm-22 RNAi worms in our study was likely a response to changes in the mitochondrial import machinery instead of a direct regulatory response to mitochondrial stress.

In a previous study conducted by Espada et al. [41] the role of ubl-5, atfs-1, and isp-1 in metformin-mediated longevity was investigated, with a primary focus on mitochondrial stress-signalling and the UPR\textsuperscript{mt}. The findings indicated that knocking down atfs-1 and isp-1 resulted in a shortened lifespan of the nematodes. Notably, when metformin was administered, the lifespan was further shortened and exacerbated metformin toxicity. However, ubl-5 did not have a significant effect and thus was reported to be required for lifespan extension. Furthermore, the study by Benedetti et al. (25) showed that ubl-5 knockdown did not induce hsp-6 and hsp-60 in response to heat stress, suggesting that ubl-5 is required for the initiation of UPR\textsuperscript{mt} and subsequent activation of chaperones.

Notably, our finding demonstrated that UPR\textsuperscript{er} can also be influenced by metformin through both inhibition and enhancement mechanisms. These findings indicate that the impact of metformin on the UPR extends beyond its effects on mitochondria. Furthermore, the effects of metformin on UPR systems may be the result of the interplay between the knockdown of target genes and their subsequent responses.

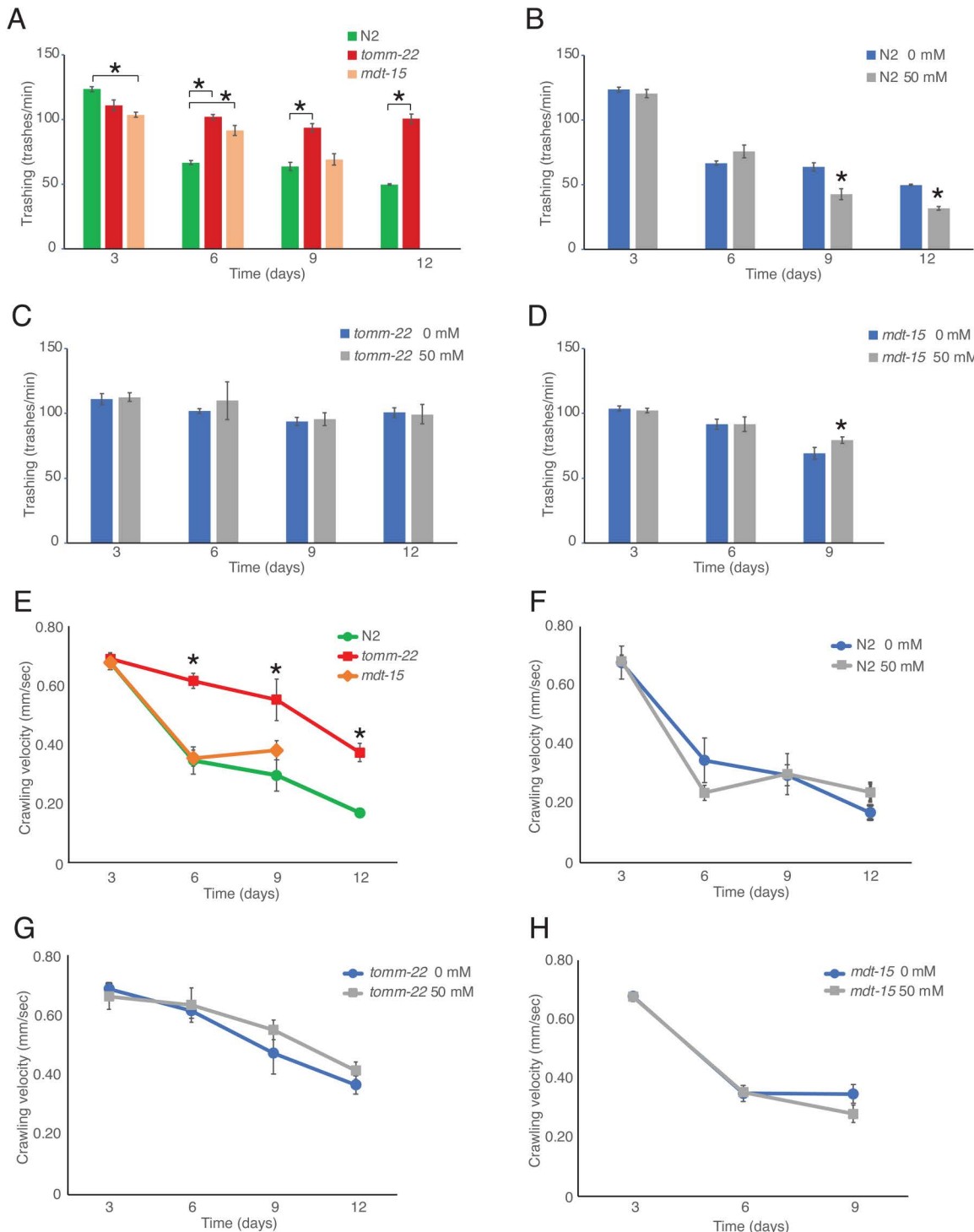

**Fig 5. Effect of metformin on locomotion in wild-type and UPR-activated *C. elegans*.** Body bending (A-D) and movement velocity (E-H) assays were performed on N2 wild-type worms fed with EV, *tomm-22* RNAi, and *mdt-15* RNAi. Worms were treated with metformin for their entire life (0 mM and 50 mM, as indicated). The assay was conducted every 3 days. Each data point represents 3 independent cultures (n = 10-20). * denotes p < 0.05.

UPRs regulate the cellular protection against various stresses, including heat and oxidative stress [50–53]. Given that many UPR proteins are heat shock proteins, the association between UPR activation and heat stress resistance is readily apparent [54]. Oxidative stress, mediated by free radicals, can disrupt protein folding, which may be counteracted by UPR responses [55]. Contrary to our expectations, the UPR$^{mt}$-activated *tomm-22* worms that gained lifespan extension by metformin did not necessarily acquire thermotolerance, while the beneficial effect of UPR$^{mt}$ activation against oxidative stress was neutralized upon metformin treatment in young adults. This effect was clear especially in wild-type worms. While young to middle-aged wild-type worms exhibited a modest increase in stress resistance upon metformin treatment, this effect rapidly diminished as they aged. In contrast, activation of the UPR$^{er}$ in *mdt-15* knockdown worms appeared to enhance thermotolerance. However, unlike wild-type worms, metformin did not further alleviate this phenotype. The oxidative stress resistance of *mdt-15* knockdown worms remained unaffected by metformin treatment, although UPR$^{er}$ was enhanced.

Mechanistically, metformin may affect cellular oxidative stress in various ways. On the one hand, metformin induces endogenous hydrogen peroxide in the worm, rendering peroxiredoxin mutants highly susceptible to metformin [8]. Conversely, metformin stimulates the oxidative stress response by upregulating nuclear accumulation of SKN-1 [6]. These multifaceted effects of metformin on cellular oxidative stress systems could complicate the longevity and stress resistance phenotypes, particularly in UPR-activated backgrounds. In the UPR$^{er}$-activated *mdt-15* worms, metformin treatment appears to induce a slight reduction in lifespan, which is concurrent with a corresponding reduction in oxidative stress resistance. However, the effect size is very small and these phenotypes may not be causally related. This is further supported by the absence of a consistent correlation between stress resistance and lifespan phenotypes in wild-type and *tomm-22* worms. Metformin extends lifespan in both wild-type and certain UPR-activated worms (*tomm-22* and *tmem-131*), but increases thermotolerance only in young wild types. Furthermore, its impact on oxidative stress resistance exhibits considerable variability and complexity over time.

We also investigated locomotory parameters including trashing and crawling velocity, as locomotion can reflect health conditions related to neuromuscular activity [56]. Activation of both UPR systems exerts a beneficial effect on locomotion. This effect was particularly striking in *tomm-22* knockdowns, as evidenced by significantly increased trashing and crawling throughout the entire tested lifespan. We hypothesized that this might result from an increase in chaperonic activity of heat shock proteins by activation of UPR$^{mt}$, which mitigates proteotoxicity that generally increases over life span. However, other mechanisms secondarily to UPR$^{mt}$ activation may also play a role in this effect. Given the short lifespan of *tomm-22*, this suggests that lifespan extension does not necessarily translate into improved health conditions, as evidenced by the healthspan metrics. Our study contributes to this observation, complementing the findings of Bansal *et al.* [57]. Notably, metformin did not appear to improve locomotion in old-aged worms. This contradicts our expectation, as it has been demonstrated that metformin can prevent sarcopenia and enhance physical performance in human subjects [58–60].

## Conclusion

Metformin extends lifespan of wild-type *C. elegans*, although its effects vary among UPR$^{mt}$-activated and UPR$^{er}$-activated worms. While metformin exerts diverse effects on the UPR systems, it can reduce UPR$^{mt}$ and UPR$^{er}$ in certain UPR-activated strains and temporarily decreases oxidative stress resistance in young worms. In some UPR, locomotor phenotypes exhibit improvement, whereas metformin treatment does not induce such changes. Therefore, metformin exhibits diverse effects on UPR, lifespan and stress resistance, contingent upon the mitochondrial and ER status of the worms.

## Supporting information

**S1 Fig. Inactivation of UPR$^{mt}$ by *atfs-1* and *ubl-1* contributes to longevity observed in metformin-treated *tomm-22*.**
(PDF)

**S1 File. Raw data for each experiment.**
(ZIP)

## Author contributions

**Conceptualization:** Jerald Tan, Bart P. Braeckman, Wichit Suthammarak.

**Data curation:** Jerald Tan, Chutipong Chiamkunakorn, Kanpapat Boonchuay, Yiying Shi, Bart P. Braeckman, Wichit Suthammarak.

**Formal analysis:** Jerald Tan, Chutipong Chiamkunakorn, Yiying Shi, Bart P. Braeckman, Wichit Suthammarak.

**Funding acquisition:** Wichit Suthammarak.

**Investigation:** Jerald Tan, Chutipong Chiamkunakorn, Kanpapat Boonchuay, Yiying Shi, Bart P. Braeckman, Wichit Suthammarak.

**Methodology:** Wichit Suthammarak.

**Project administration:** Chutipong Chiamkunakorn, Kanpapat Boonchuay.

**Validation:** Wichit Suthammarak.

**Writing – original draft:** Jerald Tan, Chutipong Chiamkunakorn, Wichit Suthammarak.

**Writing – review & editing:** Jerald Tan, Chutipong Chiamkunakorn, Bart P. Braeckman, Wichit Suthammarak.

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
