## [Decision Letter · Decision Letter 0]

Apr 15 2025

Dear Dr. Suthammarak,

We look forward to receiving your revised manuscript.

Kind regards,

David M. Ojcius

Academic Editor

PLOS ONE

Journal Requirements:

This work is supported by National Research Council of Thailand (NRCT) and Mahidol University (NRCT5-RSA63015-20). Some strains were provided by the CGC, which is funded by NIH Office of Research Infrastructure Programs (P40 OD010440). We would like to thank Brian Onken from Rutgers, The State University of New Jersey, and Profs. Phil Morgan and Margaret Sedensky from University of Washington for their valuable inputs.

- Author who received the fund: WS

- Funder and grant number: National Research Council of Thailand (NRCT) and Mahidol University (NRCT5-RSA63015-20)

- URL of funder website: https://www.nrct.go.th/

- C. elegans strains were provided by the CGC, which is funded by NIH Office of Research Infrastructure Programs (P40 OD010440).

** The funders absolutely did not play any role in the study design, data collection and analysis, decision to publish, or preparation of the manuscript**

5. We note you have included a table to which you do not refer in the text of your manuscript. Please ensure that you refer to Table 1 in your text; if accepted, production will need this reference to link the reader to the Table.

Reviewers' comments:

Reviewer's Responses to Questions

**Comments to the Author**

1. Is the manuscript technically sound, and do the data support the conclusions?

Reviewer #1: Partly

2. Has the statistical analysis been performed appropriately and rigorously?

Reviewer #1: Yes

3. Have the authors made all data underlying the findings in their manuscript fully available?

Reviewer #1: Yes

4. Is the manuscript presented in an intelligible fashion and written in standard English?

Reviewer #1: No

Reviewer #1: The manuscript by Tan and colleagues explores the effects of metformin on lifespan, stress resistance, and locomotion in C. elegans and its relationship with Unfolded Protein Response (UPRmt and UPRer) pathways. In particular, the study demonstrates differential effects on UPRmt in tomm-22 and cco-1 RNAi worms and presents a complex interaction between metformin treatment and stress responses. While the topic is timely and the study potentially offers new insights into the metformin’s action on C. elegans longevity, several aspects of the manuscript require clarification and/or further experimental support.

Major Comments:

1. The manuscript states that “metformin’s action is mitochondria‐specific” and that “UPRmt is indispensable for lifespan extension.” However, the data (especially the lifespan extension observed in both UPRmt- and UPRer-activated strains) suggest that these statements may be overgeneralized. Only two representative RNAi inducer per UPR pathway (tomm-22 and cco-1 for UPRmt; mdt-15 and tmem-131 for UPRer) has been tested. It would strengthen the study to include additional examples of UPRmt and UPRer inducers to verify whether the observed patterns are robust.

2. The authors should clarify whether they mean that the beneficial effect in tomm-22 RNAi-treated worms is due to metformin resolving mitochondrial stress or if alternative mechanisms might be at play. What do authors believe it is the mechanism behind metformin action, is a decrease on protein translation that is alleviating the stress on tomm-22 RNAi-treated worms?

3. The discussion suggests that metformin’s beneficial effects on lifespan are independent of UPRmt/UPRer activation. This conclusion partially conflicts with previous work (Espada et al., 2019/PMID: 33139960), which show that ubl-5 is required for lifespan induced by metformin in C. elegans. A discussion on how the present findings compare and contrast with this previous paper is necessary. Alternatively, authors could do the assay with RNAi against ubl-5. This would be important to support their claim that metformin action on lifespan is not dependent on UPRmt.

4. Cabreiro et al., 2013 (PMID: 23540700) also reported that metformin extends C. elegans lifespan on OP50, but not on HT115. Could authors also please provide an explanation of the apparent discrepancy between this study and Cabreiro’s work?

5. Figure 2C and 2E: The representative images of GFP fluorescence for the empty vector control appear inconsistent with the reported quantification. Moreover, a better description of how the fluorescence was analyzed would improve the manuscript. For example, in 2E are the values displayed a quantification per worm or of a whole section? Was there a normalization of the background for each image?

6. Figure 5 shows that mdt-15 RNAi-treated worms exhibit increased movement when compared to N2 despite being short-lived. This is counterintuitive. Can authors please provide an explanation for this? The manuscript would also improve from a better methodological description of how crawling velocity was obtained.

Minor Comments:

1. The introduction section would benefit from a more detailed discussion of the molecular mechanisms involved in UPRmt and UPRer activation/regulation.

2. In ‘’These findings suggest that metformin’s action is mitochondria-specific. Furthermore, they imply that UPRmt is indispensable for lifespan extension, as the metformin-induced reduction of UPRmt (Fig 2B) did not diminish its lifespan-extending effect’’. I believe the authors meant to say “dispensable” instead of ‘’indispensable’’. Please confirm the intended meaning, as the data suggest that even when UPRmt is suppressed (as in tomm-22 RNAi worms), lifespan extension still occurs.

3. The manuscript should clearly define what were the control conditions (N2/control) in all experiments. In some legend figures, it is not clear whether “control” refers to worms grown on OP50, HT115, or HT115 harboring L4440. Moreover, it was not clear what vehicle was used for metformin. Was it water? Was vehicle included on 0mM plates?

4. The quantitative data in Figure 1, Figure 2 should include statistical information on the figure rather than relying solely on the information displayed on tables.

5. There are some typos in the text and it should be double-checked for spelling (e.g., legend of figure 1 ‘’ifespan’’ instead of ‘’lifespan’’).

**Do you want your identity to be public for this peer review?** For information about this choice, including consent withdrawal, please see our Privacy Policy

Reviewer #1: No

---

## [Author Response · Author response to Decision Letter 1]

22 May 2025

Journal Requirements:

Response: We meticulously ensure that the revised manuscript conforms to the style guidelines prescribed by PLOS ONE.

Response: The previously provided information has been precisely updated upon our resubmission.

This work is supported by National Research Council of Thailand (NRCT) and Mahidol University (NRCT5-RSA63015-20). Some strains were provided by the CGC, which is funded by NIH Office of Research Infrastructure Programs (P40 OD010440). We would like to thank Brian Onken from Rutgers, The State University of New Jersey, and Profs. Phil Morgan and Margaret Sedensky from University of Washington for their valuable inputs.

- Author who received the fund: WS

- Funder and grant number: National Research Council of Thailand (NRCT) and Mahidol University (NRCT5-RSA63015-20)

- URL of funder website: https://www.nrct.go.th/

- C. elegans strains were provided by the CGC, which is funded by NIH Office of Research Infrastructure Programs (P40 OD010440).

** The funders absolutely did not play any role in the study design, data collection and analysis, decision to publish, or preparation of the manuscript**

Response: We apologize for any inconvenience caused. The funding statement provided in the initial submission contained accurate information regarding all funders. In addition, we have decided to omit the “Acknowledge” section from the revised manuscript.

Response: The requisite dataset has been uploaded as Supporting Information files in our resubmission.

5. We note you have included a table to which you do not refer in the text of your manuscript. Please ensure that you refer to Table 1 in your text; if accepted, production will need this reference to link the reader to the Table.

Response: Table 1 has been referenced in the revised manuscript’s text.

Reviewers' comments:

Reviewer's Responses to Questions

Comments to the Author

1. Is the manuscript technically sound, and do the data support the conclusions?

Reviewer #1: Partly

2. Has the statistical analysis been performed appropriately and rigorously?

Reviewer #1: Yes

3. Have the authors made all data underlying the findings in their manuscript fully available?

Reviewer #1: Yes

4. Is the manuscript presented in an intelligible fashion and written in standard English?

Reviewer #1: No

5. Review Comments to the Author

Reviewer #1: The manuscript by Tan and colleagues explores the effects of metformin on lifespan, stress resistance, and locomotion in C. elegans and its relationship with Unfolded Protein Response (UPRmt and UPRer) pathways. In particular, the study demonstrates differential effects on UPRmt in tomm-22 and cco-1 RNAi worms and presents a complex interaction between metformin treatment and stress responses. While the topic is timely and the study potentially offers new insights into the metformin’s action on C. elegans longevity, several aspects of the manuscript require clarification and/or further experimental support.

Major Comments:

1. The manuscript states that “metformin’s action is mitochondria‐specific” and that “UPRmt is indispensable for lifespan extension.” However, the data (especially the lifespan extension observed in both UPRmt- and UPRer-activated strains) suggest that these statements may be overgeneralized. Only two representative RNAi inducer per UPR pathway (tomm-22 and cco-1 for UPRmt; mdt-15 and tmem-131 for UPRer) has been tested. It would strengthen the study to include additional examples of UPRmt and UPRer inducers to verify whether the observed patterns are robust.

Response: We acknowledge your concern regarding this matter. After careful deliberation and the acquisition of additional experiments (in new Figure 2), we have modified our interpretation in the revised manuscript. To enhance the comprehensiveness of our analysis regarding the interaction between metformin and the UPRs, we also tested additional UPR-activated worms. These included nuo-6 RNAi (UPRmt) and tag-335 RNAi (UPRer) worms, whose lifespan and stress resistance phenotypes, however, were not carried on as in others. Our new data indicates that metformin exhibits the ability to affect both UPRmt- and UPRer-activated strains. This is discussed in the Discussion section of the revised manuscript.

2. The authors should clarify whether they mean that the beneficial effect in tomm-22 RNAi-treated worms is due to metformin resolving mitochondrial stress or if alternative mechanisms might be at play. What do authors believe it is the mechanism behind metformin action, is a decrease on protein translation that is alleviating the stress on tomm-22 RNAi-treated worms?

Response: We demonstrated the beneficial effects of metformin in tomm-22 RNAi-treated worms, including lifespan extension (Fig. 1C) and promoting late-life oxidative stress resistance (Fig 4B and Table 3).

In the context of lifespan alteration, the activation of UPRmt in tomm-22 worms resulted in a shortened lifespan (Fig. 1C). However, metformin demonstrated the ability to enhance the lifespan of these worms. To elucidate the underlying mechanism, We conducted a new experiment (presented in Supplementary) in which we inactivated the transcription factor and regulator of the UPRmt, ATFS-1 and UBL-5, in metformin-fed tomm-22 worms using RNAi knockdown. This experiment aimed to validate our hypothesis that the lifespan extension observed with metformin in tomm-22 worms is UPRmt-independent. The results indicated that when UPRmt is inactivated, the worms exhibited a significant lifespan extension compared to those fed metformin. This suggests that the activation of UPRmt in tomm-22 has a detrimental impact (possibly due to its other secondary effects) on the worm’s lifespan, which metformin can counteract.

Activation UPRmt in tomm-22 can theoretically activate the proteomic stress response, which should benefit the worm’s antioxidant system in mitochondria. However, the direct effect of tomm-22 knockdown is the impairment of the mitochondrial transport system, as previously addressed in the Discussion. Consequently, mitochondrial protein chaperones may not be able to function at their full capacity. Notably, metformin was shown to promote late-life oxidative stress resistance (Fig. 4B and Table 3). Therefore, metformin may exhibit additional unknown effects that mitigate the impairment of the mitochondrial transport system, facilitating the entry of protein chaperones into mitochondria and ultimately enhancing the antioxidant system.

3. The discussion suggests that metformin’s beneficial effects on lifespan are independent of UPRmt/UPRer activation. This conclusion partially conflicts with previous work ( ), which show that ubl-5 is required for lifespan induced by metformin in C. elegans. A discussion on how the present findings compare and contrast with this previous paper is necessary. Alternatively, authors could do the assay with RNAi against ubl-5. This would be important to support their claim that metformin action on lifespan is not dependent on UPRmt

Response: The previous work by Espada et al., 2020, investigated the role of ubl-5, atfs-1 and isp-1 in metformin mediated longevity and primarily focused on mitochondrial stress signaling and UPRmt. Their results on atfs-1 and isp-1 knockdown shortened lifespan of the nematodes but more importantly when treated with metformin, shortened lifespan further and exacerbated metformin toxicity. ubl-5, however, did not have a significant effect and thus is reported to be required for lifespan extension.

It is important to note that a previous study by Benedetti et al., 2006, Figure 3A, showed that ubl-5 knockdown did not induce hsp-6 and hsp-60::GFP in heat stress, suggesting that ubl-5 is required for the initiation of UPRmt and subsequent activation of chaperones. In contrast, the primary role of tomm-22 is a mitochondrial outer membrane transporter rather than a UPRmt stress regulator. The activation of hsp-6 response in our study was likely a response to changes in the mitochondrial import machinery instead of a direct regulatory response to mitochondrial stress.

As suggested by the reviewer, we did another experiment to investigate whether tomm-22 and ubl-5 have overlapping pathways by performing RNAi knockdown on both genes and treating the worms with metformin (Supplementary data, Fig. 1). Logically, if ubl-5 was required for metformin-mediated longevity, then the effect of metformin would be lost in the presence of a tomm-22 background.

We found that tomm-22 and ubl-5 double knockdown with metformin synergized and extended lifespan significantly when compared to tomm-22 treatment control. These findings suggest that both tomm-22 and ubl-5 do not overlap in their pathways but also contest the notion that ubl-5 is required for metformin-mediated longevity.

We argue that ubl-5 is not required for metformin-mediated longevity and its mitochondrial impairment effects merely exacerbate toxicity of metformin instead, similar to atfs-1 and isp-1. However, the disruption in mitochondrial import machinery ameliorated these effects (likely by disrupting the flux of chaperone proteins and mitochondrial activity) and prevented mitochondrial impairment (as described by Espada), maintaining the metformin-mediated longevity effect.

It was also important to note that Espada et al. also mentioned that they tested whether UPRmt was affected by metformin using hsp-6::GFP, and found that metformin administration did not lead to elevated expression of GFP nor reduction of mitochondrial protein levels. They have also cited De Haes et al., 2014, where the lack of UPRmt induction by metformin was reported. In this case, our findings were in line with both Espada and De Haes, as we did find that there was no UPRmt induction by metformin.

Essentially, our work highlights that metformin can extend lifespan without a strict dependency on UPRmt, as we found that cco-1 knockdown (impairing cytochrome c oxidase subunit 1) upregulated hsp-6::GFP response while also extending lifespan. The contrasting result from tomm-22 further highlights the complexity of the relationship between mitochondrial stress and lifespan extension and emphasizes that UPRmt is not an absolute requirement for metformin-mediated lifespan extension. Whether metformin-mediated lifespan extension being dependent on ubl-5 is beyond the scope of this study.

4. Cabreiro et al., 2013 (PMID: 23540700) also reported that metformin extends C. elegans lifespan on OP50, but not on HT115. Could authors also please provide an explanation of the apparent discrepancy between this study and Cabreiro’s work?

Response: It is indeed quite difficult to pinpoint the exact reason for the observed difference in lifespan results between our study and theirs, particularly because both studies were conducted under separate laboratory conditions which introduces variables that could influence the outcomes.

One possible factor that could explain the difference in lifespan would likely be the use of FUdR (5-fluoro-2’-deoxyuridine) in their experiments. In contrast to their approach, we omitted FUdR from our lifespan experiments to avoid potential confounding effects, as the impact of FUdR on lifespan has been shown to be significant.

The use of FUdR has been reported to affect gene expression patterns (McIntyre et al., 2021) and metabolic profiles of C. elegans (Davies et al., 2012). FUdR works by inhibiting DNA synthesis, which leads to reduced cell proliferation particularly in the reproductive ti

---

## [Decision Letter · Decision Letter 1]

Metformin Modulates the Unfolded Protein Responses, Altering Lifespan and Health-Promoting Effects in UPR-Activated Worms

PONE-D-25-06689R1

Dear Dr. Suthammarak,

We’re pleased to inform you that your manuscript has been judged scientifically suitable for publication and will be formally accepted for publication once it meets all outstanding technical requirements.

Kind regards,

David M. Ojcius

Academic Editor

PLOS ONE

Reviewers' comments:

Reviewer's Responses to Questions

**Comments to the Author**

Reviewer #1: All comments have been addressed

2. Is the manuscript technically sound, and do the data support the conclusions?

Reviewer #1: Yes

3. Has the statistical analysis been performed appropriately and rigorously?

Reviewer #1: Yes

4. Have the authors made all data underlying the findings in their manuscript fully available?

Reviewer #1: Yes

5. Is the manuscript presented in an intelligible fashion and written in standard English?

Reviewer #1: Yes

Reviewer #1: All of my concerns were thoroughly addressed by the authors in this round of revision, and I am now satisfied with the manuscript’s clarity, completeness, and overall quality.

**Do you want your identity to be public for this peer review?** For information about this choice, including consent withdrawal, please see our Privacy Policy

Reviewer #1: No

---

## [Editor Report · Acceptance letter]

PONE-D-25-06689R1

PLOS ONE

Dear Dr. Suthammarak,

I'm pleased to inform you that your manuscript has been deemed suitable for publication in PLOS ONE. Congratulations! Your manuscript is now being handed over to our production team.

Kind regards,

on behalf of

Dr. David M. Ojcius

Academic Editor

PLOS ONE